# Essential Involvement of Neutrophil Elastase in Acute Acetaminophen Hepatotoxicity Using BALB/c Mice

**DOI:** 10.3390/ijms24097845

**Published:** 2023-04-25

**Authors:** Yuko Ishida, Siying Zhang, Yumi Kuninaka, Akiko Ishigami, Mizuho Nosaka, Isui Harie, Akihiko Kimura, Naofumi Mukaida, Toshikazu Kondo

**Affiliations:** Department of Forensic Medicine, Wakayama Medical University, Wakayama 641-0012, Japan

**Keywords:** drug-induced hepatotoxicity, acetaminophen, neutrophils, neutrophil elastase, sivelestat, inflammation, mice

## Abstract

Intense neutrophil infiltration into the liver is a characteristic of acetaminophen-induced acute liver injury. Neutrophil elastase is released by neutrophils during inflammation. To elucidate the involvement of neutrophil elastase in acetaminophen-induced liver injury, we investigated the efficacy of a potent and specific neutrophil elastase inhibitor, sivelestat, in mice with acetaminophen-induced acute liver injury. Intraperitoneal administration of 750 mg/kg of acetaminophen caused severe liver damage, such as elevated serum transaminase levels, centrilobular hepatic necrosis, and neutrophil infiltration, with approximately 50% mortality in BALB/c mice within 48 h of administration. However, in mice treated with sivelestat 30 min after the acetaminophen challenge, all mice survived, with reduced serum transaminase elevation and diminished hepatic necrosis. In addition, mice treated with sivelestat had reduced NOS-II expression and hepatic neutrophil infiltration after the acetaminophen challenge. Furthermore, treatment with sivelestat at 3 h after the acetaminophen challenge significantly improved survival. These findings indicate a new clinical application for sivelestat in the treatment of acetaminophen-induced liver failure through mechanisms involving the regulation of neutrophil migration and NO production.

## 1. Introduction

Acetaminophen (APAP or paracetamol) is a commonly used analgesic and antipyretic drug considered safe at therapeutic concentrations. However, when overdosed, APAP can cause severe liver damage, the most frequent cause of acute liver failure (ALF) in the United States and Europe [1,2]. APAP is rapidly taken up by the intestine and processed by hepatocytes into the reactive metabolite, *N*-acetyl-*p*-benzoquinone imine (NAPQI), which causes oxidative stress and hepatocyte necrosis followed by damage-associated molecular patterns (DAMPs) release [3,4]. When activated by DAMPs, resident macrophages produce proinflammatory cytokines and chemokines, stimulating leukocyte recruitment to the liver [3,4]. The products of these cells amplify liver damage; however, they can also accelerate resolution. Therefore, to consider liver transplantation in advanced ALF, the chances of survival must be high [2,5,6]. Accordingly, novel medical therapies must be developed to prevent liver damage and its progression to liver failure. This may reduce the need for transplantation and reduce APAP-related deaths.

Recent studies have focused on the underlying immunological mechanisms of APAP-induced liver injury and their potential to regulate inflammatory responses as a therapeutic strategy [3,4,7,8,9,10,11]. A severe increase in neutrophils is characteristic of the early stages of leukocyte infiltration during APAP-induced liver injury [12,13]. Neutrophil movement is governed by responses sensed by multiple receptors, and one of the most important neutrophil responses is facilitated by CXCL8/IL-8 [14]. Human CXCL8 and its mouse analogs CXCL1, CXCL2, and CXCL5 are recognized by CXCR1 and CXCR2. Blocking the CXCR2 axis is effective in reducing acute and chronic inflammatory diseases such as drug-induced liver injury, tumor progression, and arthritis [15,16,17,18]. We have previously reported that *Cxcr2*^−/−^ mice exhibited alleviated APAP-induced liver injury with reduced neutrophil infiltration [9]. In addition, the degree of APAP-induced liver injury was well-correlated to intrahepatic neutrophil number, suggesting that neutrophil recruitment is involved in the pathogenesis of APAP-induced liver injury [10,11,19]. However, the contribution of neutrophils to the severity of liver damage is still debated [3,4].

Neutrophil elastase (NE) is an active protease released from neutrophils involved in tissue damage by inducing direct cytotoxicity and proinflammatory mediator release [20,21,22,23]. The therapeutic potential of sivelestat, an NE inhibitor, has been studied in the pathogenesis of inflammatory diseases in mice [24,25,26,27,28,29]. Recently, Yamazaki et al. investigated the effect of sivelestat on the postoperative outcomes of liver resection [30]. Treatment with sivelestat significantly reduced the inflammatory response; however, it had no protective effects on liver function after liver resection. Moreover, the combination of sivelestat and N-acetylcysteine (NAC) in mice 2 h after administration of APAP has been reported to reduce the inflammatory response and liver damage and to be more effective than NAC monotherapy [31]. However, the therapeutic effects of monotherapy with sivelestat ALF are unclear. Therefore, this study investigated the protective effect of sivelestat against APAP-induced lethal ALF in mice, and it evaluated the therapeutic potential of sivelestat in the management of fulminant liver failure.

Herein, we report that sivelestat treatment reduced hepatic leukocyte infiltration, especially neutrophils, NOS-II expression, and subsequent liver damage after APAP overdose. Furthermore, the therapeutic administration of sivelestat at 3 h after the APAP challenge significantly reduced hepatotoxicity and improved the survival rate in mice.

## 2. Results

### 2.1. Improvement of Mortality in Sivelestat-Treated Mice after APAP Challenge

To evaluate the role of NE in APAP-induced liver injury, mice were administered sivelestat or PBS 30 min after the APAP challenge. In the PBS-treated group, 50% of the mice (4 animals died within 12 h, 2 within 24 h, total of 6/12 animals died within 24 h) succumbed to acute liver injury within 24 h of the challenge. In contrast, all the sivelestat-treated mice (12/12 mice) survived for 48 h (Figure 1a). In addition, NE production was markedly increased in the liver tissues of control mice after the APAP challenge (Figure 1b). Conversely, the NE levels were significantly lower in sivelestat-treated mice than in control mice after the APAP challenge. These observations indicated that NE could contribute to the development of APAP-induced liver injury.

### 2.2. Sivelestat Treatment Attenuates Liver Damage Due to APAP Overdose

In mice treated with PBS, serum ALT levels progressively increased at 6 h and reached a maximum level (19,950.0 ± 1417.2 IU/L) at 10 h after the APAP challenge. However, the elevation of serum ALT levels in sivelestat-treated mice was markedly reduced (10 h, 1180 ± 300.8 IU/L) compared with that in PBS-treated mice (Figure 2a). Similarly to ALT levels, serum AST levels progressively increased at 6 h and reached a maximum level (5786.7 ± 2326.4 IU/L) at 10 h after the APAP challenge (Figure 2b). Histopathologically, at 6 h after the APAP challenge, severe hemorrhages and centrilobular hepatic necrosis were observed in PBS-treated mice. These histopathological changes were still evident 24 h after the APAP challenge (Figure 2c). By contrast, these morphological alterations were diminished in sivelestat-treated mice (Figure 2c,d), which may result in the reduction of serum ALT and AST levels.

### 2.3. Treatment with Sivelestat Reduces Hepatic Neutrophil Infiltration after APAP Challenge

Several lines of accumulating evidence have suggested that leukocyte recruitment is essential for the development of APAP-induced hepatotoxicity [3,32,33]. Therefore, we evaluated the influence of sivelestat on intrahepatic leukocyte recruitment after the APAP challenge. The number of neutrophils in the liver of control mice progressively increased at 6 h after the APAP challenge and remained elevated at 24 h. By contrast, intrahepatic infiltration of neutrophils was significantly reduced at 6 and 24 h after the APAP challenge in sivelestat-treated mice, compared with that in PBS-treated mice (Figure 3a,b). In line with this, sivelestat treatment significantly alleviated intrahepatic MPO activity after the APAP challenge (Figure 3c). However, both F4/80-positive macrophages and CD3-positive T cells were recruited to a similar extent into the livers of sivelestat-treated and PBS-treated mice (Figure 4a–d). These observations suggested that the NE inhibitor alleviated intrahepatic neutrophil accumulation in APAP-induced acute liver injury.

### 2.4. Attenuated Intrahepatic Gene Expression of Inflammatory Cytokines and Chemokines in Sivelestat-Treated Mice after APAP Challenge

Several lines of accumulating evidence have suggested that inflammatory cytokines and chemokines play detrimental roles in APAP-induced liver injury [3,9,11,34]. For example, the APAP challenge increased the expressions of *Ifng*, *Tnfa*, *Il1b*, *Cxcl1*, and *Cxcl2* in the livers of PBS-treated mice after the APAP challenge (Figure 5). However, in sivelestat-treated mice, the increased gene expression was significantly reduced at 6 and/or 24 h of the APAP challenge compared to those in PBS-treated mice. These observations suggested that the inhibition of NE reduced the expression of inflammatory cytokines and chemokines and subsequent inflammatory responses in the liver after the APAP challenge.

### 2.5. Sivelestat Treatment Reduced Hepatic Nos2 Expression and NO Production after APAP Challenge

Next, we examined the intrahepatic expression of *Nos2* (gene expression of NOS-II) and nitrotyrosine (a marker of peroxynitrite-induced tissue injury). *Nos2* expression was significantly increased in control mice 24 h after the APAP challenge but not in the sivelestat-treated mice (Figure 6a). Consistent with *Nos2* expression, nitrotyrosine was intensively detected in the centrilobular regions of the livers of PBS-treated but not sivelestat-treated mice 24 h after the APAP challenge (Figure 6b).

### 2.6. Gene Expression of Inflammatory Cytokines and CXC Chemokines in NE-Treated Macrophages

Macrophages are the main producers of inflammatory cytokines and chemokines during APAP-induced liver injury. To understand the role of NE in inducing inflammatory cytokine and CXC chemokine expression, mice macrophages from peritoneal exudate cells (PECs) were exposed to purified NE, and gene expressions for IFN-γ, TNF-α, IL-1β, CXCL1, and CXCL2 were evaluated by real-time RT-PCR. The expression of all examined genes was found to be significantly increased in the NE-treated macrophages in vitro (Figure 7 and Appendix A). These results support in vivo observations and indicate that NE can mediate the induction of expression of IFN-γ, TNF-α, IL-1β, CXCL1, and CXCL2 in macrophages.

### 2.7. Sivelestat Therapy Is Effective in Improving Mortality Due to APAP Overdose in Mice

To evaluate the therapeutic effect of the NE inhibitor on mortality due to APAP overdose, mice were treated with sivelestat at 3 or 5 h after the APAP challenge. In mice treated with sivelestat at 3 h after APAP administration, the mortality was significantly improved compared with that in the control mice (Figure 8a). By contrast, mice treated with sivelestat at 5 h after the APAP challenge showed marginal but not significant improvement in mortality (Figure 8b). On the other hand, histopathological alterations, such as severe hemorrhage and centrilobular hepatic necrosis, were diminished in sivelestat-treated mice at both 3 and 5 h of the APAP challenge (Figure 8c–f).

## 3. Discussion

APAP is one of the most common over-the-counter drugs and is generally safe at the recommended therapeutic dose. However, APAP overdose is a clinical problem that can cause acute liver damage, resulting in fatal events. The current understanding of APAP-induced liver injury is that it promotes severe tissue damage owing to a combination of toxic metabolite by-products, NAPQI, and the acute inflammatory response from hepatic necrosis [35,36]. Therefore, drug-induced liver injury is a multifactorial scenario in which hepatocyte death, cytokines, and chemokines, along with resident and peripheral immune cells, promote the consequences of drug stressors [37]. This environment induces a large number of neutrophils in the liver, which significantly contributes to the progression and severity of APAP-induced hepatocellular death. For instance, since neutrophil depletion by antibody administration and CXCR2 gene deficiency alleviated tissue damage after the APAP challenge, the suppression of neutrophil migration was considered a major candidate for the treatment of APAP-induced liver injury [9,19,38,39,40].

Neutrophils are among the first immune cells to infiltrate the liver and are induced by chemokines during APAP-induced liver injury [15,38,41,42]. In recent studies, infiltrative neutrophils upregulated TLR9, a receptor that senses extracellular DNA [15], and TLR9^−/−^ mice showed diminished inflammation during APAP-induced liver injury; however, adaptive transplantation of WT neutrophils into TLR9^−/−^ mice reversed this hepatoprotective effect [38]. In addition, neutrophils may directly mediate hepatocyte necrosis in APAP-induced liver injury [15]. Thus, neutrophil infiltration is a characteristic immune response of APAP-induced liver injury [3,13]. Several lines of accumulating evidence have demonstrated that IFN-γ^−/−^ mice and NK and NKT cell-depleted mice show reduced intrahepatic neutrophil recruitment and subsequent less liver damage after APAP overdose [11,12]. Pretreatment with neutrophil antisera significantly attenuates APAP hepatotoxicity in rats [43]. In addition, Liu et al. showed that 24 h pretreatment with an anti-Gr-1 antibody induced neutropenia and diminished liver neutrophil accumulation and liver damage after APAP challenge [19]. Thus, the presence of activated intrahepatic neutrophils would exacerbate liver damage. Neutrophils are activated by inflammatory mediators, including CXC chemokines. We have previously observed that Cxcr2^−/−^ mice have reduced neutrophil infiltration and NOS-II expression in the liver compared to WT mice, which eventually results in reduced APAP-induced hepatotoxicity [9]. The present study also demonstrated the pathogenic roles of neutrophils in APAP-induced liver injury.

NE is an active protease secreted by neutrophils, and its circulating levels and activity are increased during diet-induced obesity and insulin resistance in mice [44,45]. In addition, Chen et al. found that Western diet-induced NASH mice had significantly increased expression of NE protein in the liver, and NE-deficient mice experienced recovery from NASH [21]. Our study also found a significant increase in hepatic NE after the APAP challenge. These studies suggest that the elevation of NE levels could be attributable to the excessive infiltration of neutrophils into the liver and that NE may play an important role in liver inflammation. In fact, treatment with an NE inhibitor, sivelestat, reduced neutrophil infiltration in the liver after APAP challenge but not macrophage or CD3^+^ T cell infiltration.

NE inhibitors suppress inflammatory cytokines and chemokines and reduce tissue damage [46,47,48,49,50,51]. We previously reported that IFN-γ^−/−^ and TNF-Rp55^−/−^ mice exhibited alleviated liver injury along with decreased expression of hepatic proinflammatory cytokines and chemokines and neutrophil infiltration after APAP overdose [10,11]. Consistent with these observations, several studies have shown the protective effects of NE inhibitors on ischemia-reperfusion injury with reduced expression of inflammatory cytokines and chemokines and/or diminished leukocyte infiltration [46,47,52]. In this study, sivelestat suppressed the expression of inflammatory cytokines, *Ifng*, *Tnfa*, and *Il1b,* and CXC chemokines, *Cxcl1* and *Cxcl2*, subsequent neutrophil infiltration into the liver after APAP challenge. Furthermore, NE treatment enhanced the expressions of *Ifng*, *Tnfa*, *Il1b*, *Cxcl1*, and *Cxcl2* in macrophages in vitro. Therefore, NE plays a detrimental role in APAP-induced liver injury by promoting excessive inflammation. In other words, the capacity of sivelestat to suppress the gene expressions of inflammatory cytokines and CXC chemokines may reduce neutrophil infiltration into the liver after an APAP challenge.

Nitric oxide (NO) contributes to tissue damage in various types of liver damage [53,54,55,56]. We have previously provided conclusive evidence for the involvement of NOS-II expression and subsequent NO production in APAP-induced liver injury [10,11]. In addition, we found that hepatic neutrophils are a major source of NOS-II and that neutrophil depletion reduces APAP-induced NOS-II expression and neutrophil infiltration [9]. Several lines of evidence have indicated that NOS-II regulates CXC chemokine production and has a regulatory effect on neutrophil accumulation in various acute inflammations [57,58,59,60]. Moreover, IFN-γ can enhance NOS-II expression in various cells [61,62,63,64]. We found that the expression of *Cxcl1*, *Cxcl2*, and *Ifng* was significantly enhanced in the liver after the APAP challenge. Therefore, NO generation from neutrophil NOS-II is an important autocrine regulator of neutrophil infiltration in the liver after APAP overdose.

NAC administration is the only pharmacological strategy available for patients with APAP-induced liver injury [1,65,66]. However, given the inadequacy of the current NAC treatments, new therapies that target inflammation after liver injury must be developed. Raevens et al. have reported that the combination of NAC and sivelestat suppressed the inflammatory response and reduced liver damage following 300 mg/kg APAP administration in C57BL/6 mice [31]. In this study, the survival rate of BALB/c mice administered 750 mg/kg APAP was approximately 50%, but it was observed that all survived with only sivelestat treatment. We previously demonstrated that 750 mg/kg APAP is a lethal dose for BALB/c mice, as approximately 50% of the mice died after the administration of 750 mg/kg APAP [10]. Therefore, discrepancies may be explained by differences in the mouse strains used and/or APAP doses.

Masubuchi et al. reported that Th1-dominant C57BL/6 mice were more susceptible to APAP-induced liver injury than BALB/c ones [67]. The strain difference was not attributable to hepatic drug metabolism to generate APAP-reactive metabolites because there was no difference in GSH consumption, an index of NAPQI generation, or expression of P450 enzymes involved in NAPQI generation. In addition, several studies have found strain differences in susceptibility to APAP hepatotoxicity, which was linked to some pathophysiological events in the toxicity [68,69,70]. In line with our previous studies, inflammatory cytokines, such as IFN-γ, TNF-α, IL-1β, and IL-6, and chemokines, such as CCL2, CCL3, CXCL1, and CXCL2, were significantly upregulated after APAP administration in BALB/c mice. In IFN-γ^−/−^ and TNF-Rp55^−/−^ mice (BALB/c background) exhibited significantly reduced expression of inflammatory cytokines and chemokines and subsequently reduced leukocyte infiltration [10,11]. Sivelestat administration in BALB/c mice 3 h after the APAP challenge significantly increased survival up to 48 h. Although there were no statistical differences (*p* = 0.0572), sivelestat treatment in mice 5 h after APAP challenge markedly increased their survival. These findings suggested that treatment with sivelestat would be effective within a few hours after an APAP overdose and that delaying sivelestat administration would reduce its therapeutic effects. Collectively, these findings indicate that treatment with sivelestat alone has a sufficient therapeutic effect on APAP-induced liver damage caused by APAP.

## 4. Materials and Methods

### 4.1. Reagents and Antibodies (Abs)

APAP was obtained from the Sigma Chemical Company (St. Louis, MO, USA). Sivelestat (ONO-5046) and purified NE were purchased from ONO Pharmaceutical Co., Ltd. (Osaka, Japan) and Enzo Life Sciences (Farmingdale, NY, USA), respectively. For immunohistochemical analysis, the following monoclonal antibodies (mAbs) or polyclonal antibodies (pAbs) were used; rat anti-mouse F4/80 mAb (clone BM8, Biomedicals AG, Beringen, Switzerland), rabbit anti-myeloperoxidase (MPO) pAb (Neomarkers, Fremont, CA, USA), rat anti-human CD3 mAb cross-reacting with mouse CD3 (clone CD3-12, AbD Serotec, Raleigh, NC, USA), and rabbit anti-nitrotyrosine pAb (Upstate, St. Louis, MO, USA).

### 4.2. Mice

Pathogen-free 8–12-week-old male BALB/c mice were obtained from Clea Japan, Inc. (Tokyo, Japan). All animal experiments were approved by the Committee on Animal Care and Use at Wakayama Medical University (approval number 1040), and all procedures were performed in accordance with the relevant guidelines and regulations.

### 4.3. APAP-Induced Liver Injury

APAP-induced liver injury was developed as previously described [10,11]. Briefly, APAP solution was prepared immediately before each experiment by dissolving it in PBS followed by warming it to 37 °C. In all experiments, mice had free access to water and food, although food was removed 12 h before APAP challenge. The BALB/c mice were administered 750 mg/kg APAP intraperitoneally as described previously [9,10,11]. Sivelestat at 200 mg/kg (sivelestat was dissolved in PBS) [31,71] or PBS as control was administered intraperitoneally at 30 min after the APAP administration. In another series of experiments, mice were intraperitoneally injected with sivelestat or PBS 3 or 5 h after the APAP challenge.

### 4.4. Determination of Serum Alanine Aminotransferase (ALT) and Aspartate Aminotransferase (AST) Levels

Whole blood samples were collected at the indicated time intervals after APAP injection, and those serum ALT and AST levels were measured using a Fuji DRI-CHEM 7000V (Fuji Medical System, Tokyo, Japan) as instructed by the manufacturer.

### 4.5. Histopathological Analysis

The liver tissues were collected from the mice at the indicated time intervals after APAP challenge and were immediately fixed in 4% formaldehyde buffered with PBS. Paraffin-embedded sections (6-μm thickness) were stained with hematoxylin and eosin (HE). Histopathological findings in the livers were scored as previously described [9]. Briefly, normal sections were graded 0. Minimal centrilobular necrosis was graded as 1+, more extensive necrosis confined to centrilobular regions was graded 2+, necrosis extending from the central zone to portal triads was graded 3+, and massive necrosis of most of the liver was graded 4+. All evaluations were performed by an examiner without prior knowledge of the experimental procedures.

### 4.6. Immunohistochemical Analysis

Immunohistochemical analyses were performed as previously described [9]. Briefly, the deparaffinized sections were immersed in 0.3% H_2_O_2_ in methanol for 30 min to eliminate endogenous peroxidase activities. The sections were further incubated with PBS containing 1% normal serum derived from the same species as that used for the preparation of the secondary Ab and 1% BSA to reduce nonspecific reactions. The sections were incubated with anti-MPO pAbs or anti-F4/80 mAb at a concentration of 1 µg/mL at 4 °C overnight. After the incubation with biotinylated secondary Abs (2 µg/mL) at room temperature for 30 min, immune complexes were visualized using a Catalyzed Signal Amplification System (Dako, Kyoto, Japan) according to the manufacturer’s instructions. The number of infiltrating neutrophils and macrophages in the liver was enumerated on five randomly chosen visual fields at ×400 (0.2 mm × 0.3 mm), and the average of the five selected microscopic fields was calculated. All measurements were performed by an examiner without prior knowledge of the experimental procedures.

### 4.7. ELISA

Intrahepatic NE levels were quantified using a commercial ELISA kit (ab252356, Abcam, Waltham, MA, USA) according to the manufacturer’s instructions. The detection limit was 62.5 pg/mL to 4000 pg/mL. The total protein in the supernatant was measured using a commercial kit (BCA Protein Assay Kit; Pierce, Waltham, MA, USA). Data from the tissue samples were normalized to the total protein content.

### 4.8. Myeloperoxidase Activity Assay

Myeloperoxidase (MPO) activity was measured to evaluate neutrophil recruitment using an MPO activity assay kit (ab105136, Abcam) according to the manufacturer’s instructions. The data are expressed as units per milligram of total protein.

### 4.9. Quantitative RT-PCR Analysis

Total RNA was extracted from the liver tissue using ISOGEN (Nippon Gene, Toyama, Japan), according to the manufacturer’s instructions. Next, 3 μg of total RNA was reverse transcribed to cDNA with Oligo(dT)_15_ primers using PrimeScript™ Reverse Transcriptase (Takara Bio, Shiga, Japan). The resultant cDNA was subjected to real-time PCR by using SYBR^®^ Premix Ex Taq™ II (Takara Bio) and specific primer sets (Takara Bio), as described previously [72] (Table 1). Amplification and detection of mRNA were conducted by using the Thermal Cycler Dice^®^ Real-Time System (Takara Bio, TP800), according to the manufacturer’s instructions. To standardize the mRNA concentrations, transcript levels of β-actin were determined in parallel for each sample, and relative transcript levels were normalized based on β-actin transcript levels.

### 4.10. In Vitro Culture and Stimulation of Macrophages

After an abdomen massage for 30 s, peritoneal exudate cells (PECs) were harvested from the peritoneal fluid; lavage was performed (cold PBS), and then centrifugation at 400× *g* for 5 min at 4 °C. The PEC pellet was suspended in RPMI-1640 medium supplemented with 10% FBS and incubated in a 5% CO_2_ atmosphere at 37 °C for attachment of macrophages for 2 h. Prior to NE treatment, the cells were washed twice with 1 mL of PBS and cultured for 1 h with the corresponding serum-free medium. Cells were treated with 0.5 μg/mL NE for 16 h, washed twice with 1 mL of PBS, and cultured at 37 °C in 5% CO_2_ in fresh serum-free medium for 6 h before harvesting.

### 4.11. Statistical Analysis

Data are expressed as the mean ± SEM. For the comparison between the control and NE inhibitor groups at multiple time points, 2-way ANOVA followed by Dunnett’s post-hoc test was used. An unpaired Student’s *t*-test was performed to compare the values between the two groups. Furthermore, *p* < 0.05 was considered statistically significant. All statistical analyses were performed using the Statcel3 software. The Kaplan–Meier survival curve was analyzed using a log-rank test.

## 5. Conclusions

Our results demonstrate that the NE inhibitor, sivelestat, has potent anti-inflammatory effects by decreasing neutrophil infiltration and NO production, as well as reducing cytokine and chemokine expression, and can be used as a therapeutic agent for APAP-induced liver injury. Therefore, optimizing the use of sivelestat may provide the tool that is highly needed in the treatment of liver damage.

## Figures and Tables

**Figure 1 ijms-24-07845-f001:**
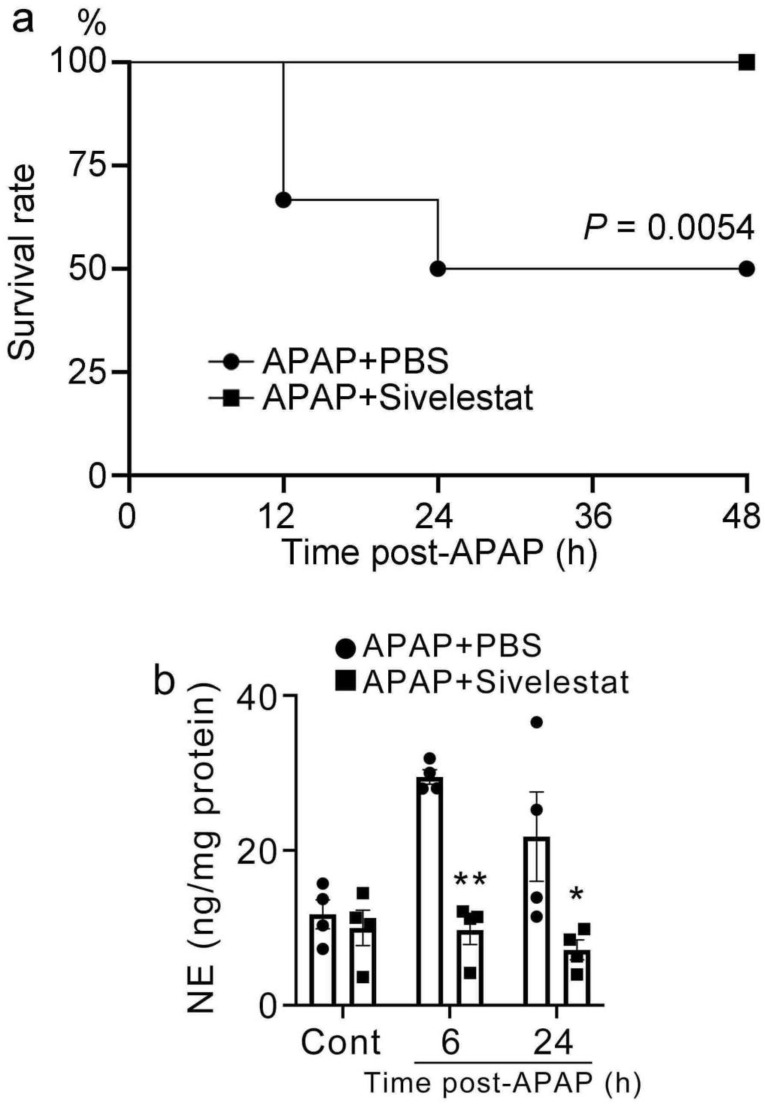
Survival rate of experimental APAP overdose and NE production in the liver after APAP challenge. (**a**) Survival rate of sivelestat-treated (n = 12) and PBS-treated control mice (n = 12) 30 min after administration of 750 mg/kg APAP. (**b**) The hepatic NE levels in mice treated with sivelestat or PBS 30 min after APAP challenge. All values represent means ± SEM (n = 4). * *p* < 0.05; ** *p* < 0.01, vs. PBS-treated control mice.

**Figure 2 ijms-24-07845-f002:**
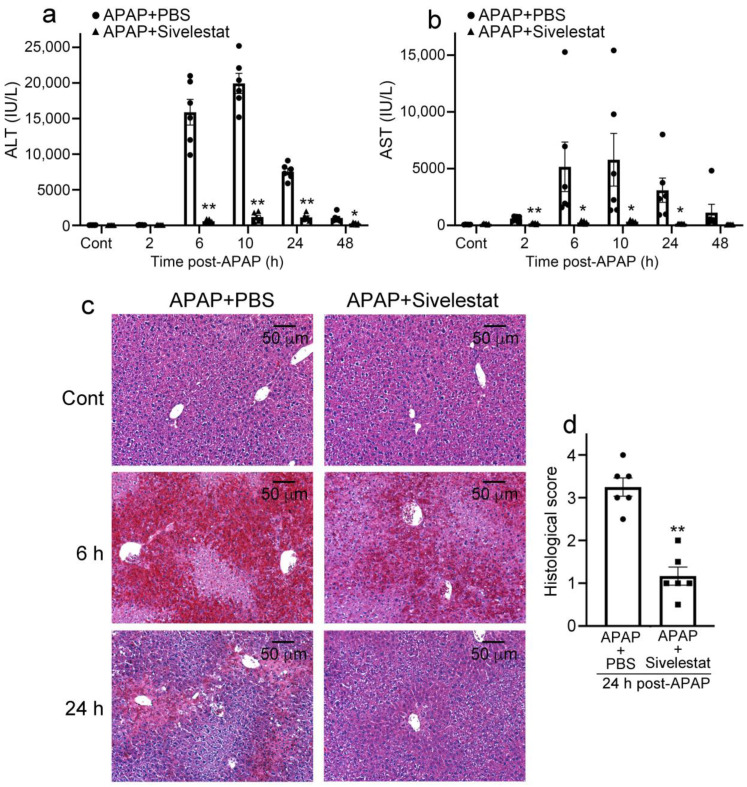
Sivelestat reduces liver damage in mice with APAP-induced acute liver injury. (**a**,**b**) Analysis on serum ALT (**a**) and AST (**b**) levels in sivelestat- and PBS-treated mice after APAP challenge. All values represent means ± SEM (n = 6). * *p* < 0.05; ** *p* < 0.01, vs. PBS-treated control mice. (**c**) Histopathological analysis on the livers from sivelestat-treated and control mice after APAP challenge (HE staining). Representative results from 6 animals at each time point are shown here. (**d**) Histological scores of the liver damage in sivelestat-treated and control mice at 24 h after APAP challenge (n = 6). All values represent means ± SEM (n = 6). ** *p* < 0.01 vs. PBS-treated control mice.

**Figure 3 ijms-24-07845-f003:**
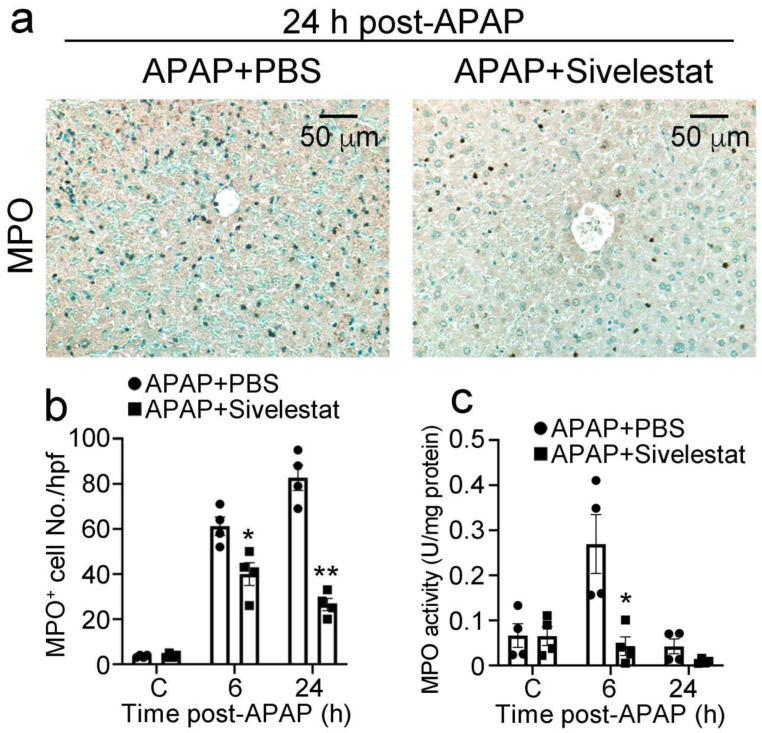
The influence of NE inhibitor on intrahepatic neutrophil recruitment after APAP challenge. (**a**) Immunohistochemical identification of neutrophils in the liver from sivelestat-treated and control mice at 24 h after APAP challenge. Representative results from 6 animals at each time point are shown. (**b**) The number of neutrophils in the livers was determined after APAP challenge. All values represent means ± SEM (n = 4). * *p* < 0.05; ** *p* < 0.01, vs. PBS-treated control mice. (**c**) Intrahepatic MPO activity was determined as described in Materials and Methods. All values represent means ± SEM (n = 4). * *p* < 0.05 vs. PBS-treated control mice.

**Figure 4 ijms-24-07845-f004:**
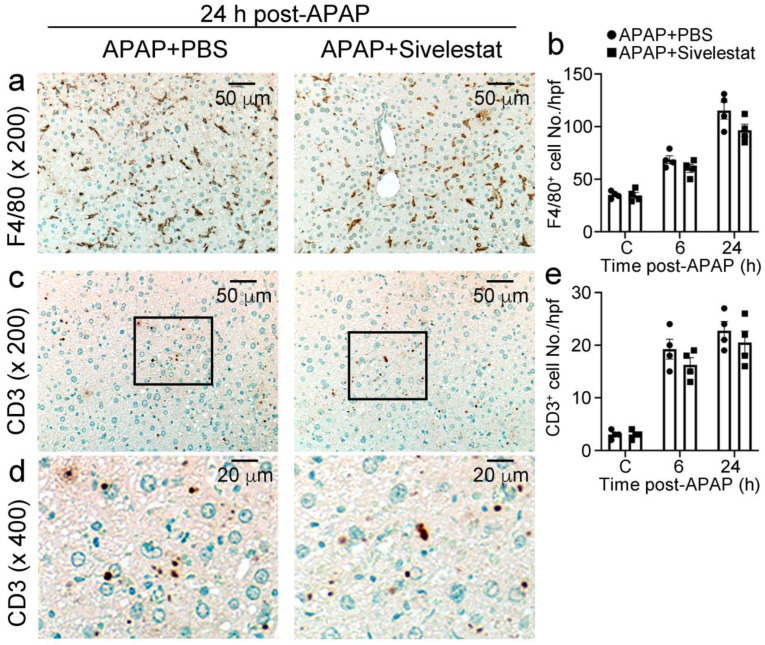
Immunohistochemical identification of macrophages and CD3^+^ T cells in the liver from sivelestat-treated and control mice at 24 h after APAP challenge. The liver tissues were obtained from each mouse and processed for an immunohistochemical analysis using anti-F4/80 and anti-CD3 antibodies to macrophages (**a**) and CD3^+^ T cells ((**c**), ×200 and (**d**), ×400 in the frames), respectively. Representative results from 6 animals at each time point are shown. The numbers of macrophages (**b**) and CD3^+^ T cells (**e**) in the livers were determined after APAP challenge. All values represent means ± SEM (n = 4).

**Figure 5 ijms-24-07845-f005:**
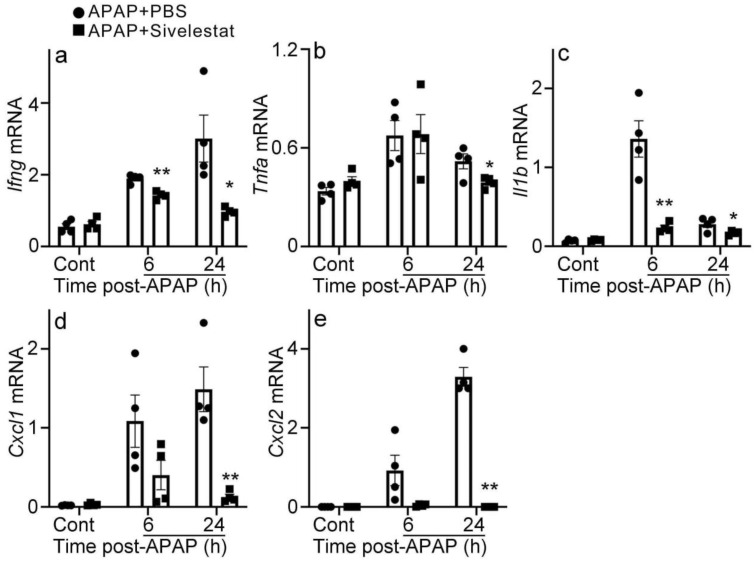
The gene expression of inflammatory cytokines and chemokines in the liver. *Ifng* (**a**), *Tnfa* (**b**), *Il1b* (**c**), *Cxcl1* (**d**), and *Cxcl2* (**e**). All values represent means ± SEM (n = 4). * *p* < 0.05; ** *p* < 0.01, vs. PBS-treated control mice.

**Figure 6 ijms-24-07845-f006:**
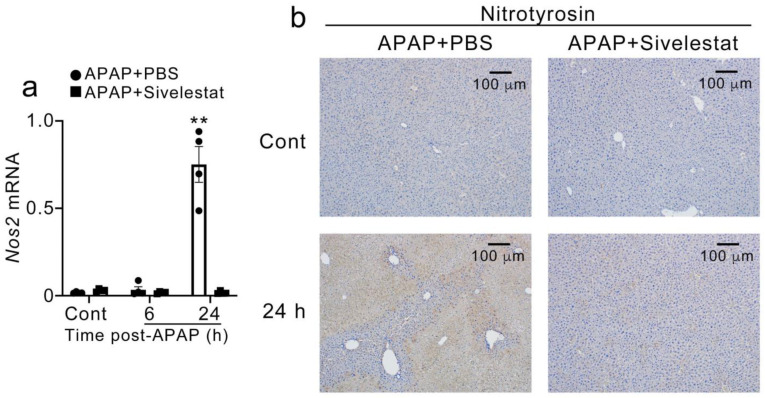
The hepatic expression of *Nos2* gene and nitrotyrosine protein after APAP challenge. (**a**) Real-time RT-PCR was performed on total RNA extracted from the liver at indicated time intervals. All values represent means ± SEM (n = 4). ** *p* < 0.01 vs. PBS-treated control mice. (**b**) Immunohistochemical analysis for nitrotyrosine. Representative results from 6 animals at each time point are shown.

**Figure 7 ijms-24-07845-f007:**
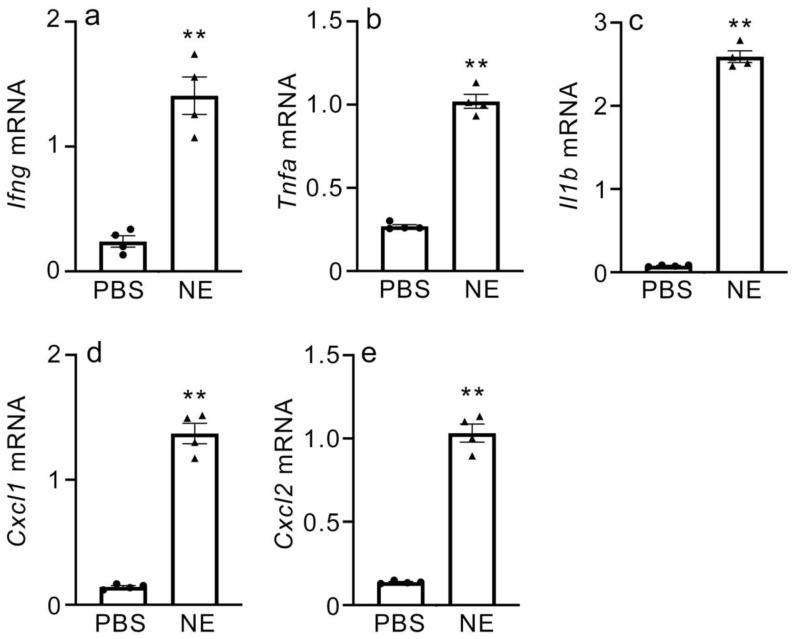
Effect of NE on expression of inflammatory cytokines and CXC chemokines in mouse macrophages from peritoneal exudate cells (PECs). *Ifng* (**a**), *Tnfa* (**b**), *Il1b* (**c**), *Cxcl1* (**d**), and *Cxcl2* (**e**). All values represent means ± SEM (four independent experiments). ** *p* < 0.01 vs. PBS treatment.

**Figure 8 ijms-24-07845-f008:**
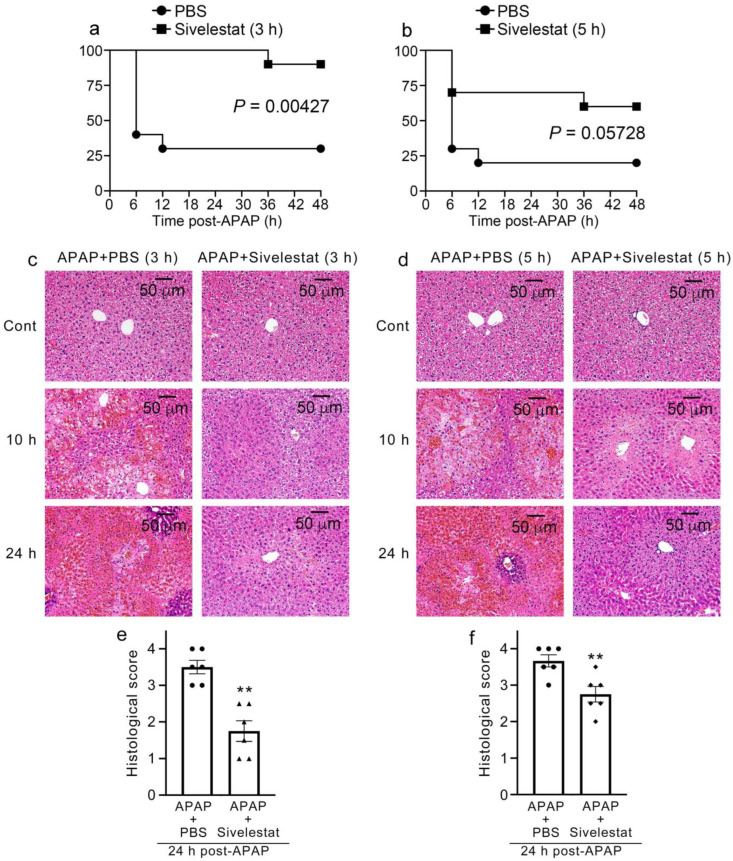
Survival rate on sivelestat therapy for APAP overdose in mice. (**a**) Survival rate of sivelestat-treated and PBS-treated control mice 3 h (**a**) or 5 h (**b**) after APAP administration (n = 10). (**c**,**d**) Histopathological analysis on the livers from sivelestat-treated 3 h (**c**) and 5 h (**d**) after APAP administration and control mice at 10 and 24 h after APAP challenge (HE staining). Representative results from 6 animals at each time point are shown here. (**e**,**f**) Histological scores of the liver damage in sivelestat-treated mice 3 h (**e**) and 5 h (**f**) after APAP challenge and control mice at 24 h after APAP challenge (n = 6). All values represent means ± SEM. ** *p* < 0.01 vs. PBS-treated control mice.

**Table 1 ijms-24-07845-t001:** Sequences of primers used for real-time RT-PCR.

Transcript	Sequence
*Ifng*	(F) 5′-CGGCACAGTCATTGAAAGCCTA-3′ (R) 5′-GTTGCTGATGGCCTGATTGTC-3′
*Tnf*	(F) 5′-AAGCCTGTAGCCCACGTCGTA-3′ (R) 5′-GGCACCACTAGTTGGTTGTCTTTG-3′
*Il1b*	(F) 5′-TCCAGGATGAGGACATGAGCAC-3′ (R) 5′-GAACGTCACACACCAGCAGGTTA-3′
*Cxcl1*	(F) 5′-TGCACCCAAACCGAAGTC-3′ (R) 5′-GTCAGAAGCCAGCGTTCACC-3′
*Cxcl2*	(F) 5’-AAAGTTTGCCTTGACCCTGAA-3’ (R) 5’-CTCAGACAGCGAGGCACATC-3’
*Actb*	(F) 5′-CATCCGTAAAGACCTCTATGCCAAC-3′ (R) 5′-ATGGAGCCACCGATCCACA-3′

(F)—Forward primer; (R)—Reverse primer.

## Data Availability

The raw data supporting the conclusions of this article will be made available by the corresponding authors (Yuko Ishida and Toshikazu Kondo) without undue reservation.

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
