# Peer review of "Essential Involvement of Neutrophil Elastase in Acute Acetaminophen Hepatotoxicity Using BALB/c Mice"

_ijms, 2023, doi:10.3390/ijms24097845_

Round 1

Reviewer 1 Report

The work was written well. In the manuscript, some grammatical corrections are required. After considering the corrections, the manuscript may be considered for publication.

Author Response

Reviewer 1

The work was written well. In the manuscript, some grammatical corrections are required. After considering the corrections, the manuscript may be considered for publication.

In accordance with the comments, our manuscript was carefully edited in English by an English speaker (please see the attached certification).

Reviewer 2 Report

In the present manuscript, the authors report that Sivelestat, an inhibitor of elastase, reduces hepatic leukocyte infiltration of neutrophils, the NOS-II expression, and subsequently liver damage in an acute APAP injury model in mice. The study is well conducted but requires some clarifications:

1-      Most of the studies done recently on Sivelastat and many references cites by the authors were conducted in C57BL/6 mice where Th1 immune response is prevalent whereas Th2 immunity is prevalent in Balb/c. This includes many references cited by the authors, for example references 15, 19 21, 24, 31, to cite some. Could the author comment on this and include this in the discussion? Could this affect the inflammatory if C57BL/6 were used?

2-      Raw 264.7 cells, although used by some authors to mimic macrophages, are a cell line that is usually replaced by primary macrophages whenever it is possible. In the present study, the author should use Kupffer cells as a best choice or at least primary cells such as bone-marrow derived macrophages or elicited peritoneal macrophages. The results might be different and probably some effects clearer. In figure 5 for example, the effect on INFgamma and TNFa is borderline and might be more relevant in primary cells.

3-      Figure 4 (c and d) needs higher magnification or inserts. The labeling of CD3 is not clear.

4- 

 Minor comments:

1- put clear labeling for the figures (figure 2. a, b c not clearly identifiable on the figure; same for figure 3, 4 and 6)

2-Scale bars:  to be added on all IHC figures for uniformity

3- Statistics for figures 1b (24 hours); figure 7 (b) need to be rechecked.

4- Replace iNOS by NOS-II

Author Response

Reviewer 2

1-      Most of the studies done recently on Sivelastat and many references cites by the authors were conducted in C57BL/6 mice where Th1 immune response is prevalent whereas Th2 immunity is prevalent in Balb/c. This includes many references cited by the authors, for example references 15, 19 21, 24, 31, to cite some. Could the author comment on this and include this in the discussion? Could this affect the inflammatory if C57BL/6 were used?

We have published several literatures on acetaminophen (APAP)-induced liver injury in BALB/c mice (PMID: 12153990, 14557383, 16552707). In accordance with the comments, we modified Discussion section in the revised version (line 243-254).

2-      Raw 264.7 cells, although used by some authors to mimic macrophages, are a cell line that is usually replaced by primary macrophages whenever it is possible. In the present study, the author should use Kupffer cells as a best choice or at least primary cells such as bone-marrow derived macrophages or elicited peritoneal macrophages. The results might be different and probably some effects clearer. In figure 5 for example, the effect on INFgamma and TNFa is borderline and might be more relevant in primary cells.

In accordance with the comments, we performed in vitro culture experiments using macrophages from peritoneal exudate cells (PECs). The results in Figure 7 (using Raw 264.7 cells) were replaced with new results using PECs and the Raw 264.7 cells data were moved to Supplementary figure 1. In addition, the Results (lines 141-142), Materials and Methods (lines 339-346), and Figure legends (line 661) sections were revised.

3-      Figure 4 (c and d) needs higher magnification or inserts. The labeling of CD3 is not clear.

In accordance with the comments, a higher magnification figures for CD3+ cells were added (new Figure 4d and line 645).

 Minor comments:

1- put clear labeling for the figures (figure 2. a, b c not clearly identifiable on the figure; same for figure 3, 4 and 6)

In accordance with the comments, we have modified the labels on Figures 2, 3, 4, and 6.

2-Scale bars:  to be added on all IHC figures for uniformity

In accordance with the comments, scale bars were added for all IHC data in Figures 6b and 8, c and d.

3- Statistics for figures 1b (24 hours); figure 7 (b) need to be rechecked.

In accordance with the comments, we rechecked the statistical differences in Figure 1b (24 h) and Figure 7b (new Supplemental figure 1). The differences were statistically significant as p = 0.048 and 0.002, respectively. Please see the attached Excel raw data.

4- Replace iNOS by NOS-II

In accordance with the comments, iNOS were replaced with NOS-II.

Reviewer 3 Report

The manuscript by Zhang et al provides an investigation of the effects of sivelestat, an elastase inhibitor, on the severity of liver injury induced by APAP overdose in mice. The following concerns exist:

1       The first and most important is novelty. Many research groups have already used sivelestat in models of liver injury, including APAP-induced injury. The paper by Raevens et al 2020 (https://pubmed.ncbi.nlm.nih.gov/31841237/) even shows a mild but significant beneficial effect of sivelestat on APAP-induced liver injury in mice when given 2h after APAP.

2.       A second and also critical flaw is the regimen by which mice were given APAP and sivelestat. APAP was given at an extremely high dose (750/mg/kg), causing in some instances 75% mortality. This is excessive and likely leads to confusing data. Moreover, sivelestat was given very early to the mice (30 min after APAP). Although this leads to improvement in the mice condition, it is most unlikely that this has any connection to elastase inhibition, since neutrophils will only migrate to the liver several hours after the treatment. In particular, it is unexplained why there is almost complete protection at 6 h after APAP when neutrophils are just beginning to accumulate in the liver. This all points to off-target effects of sivelestat that have nothing to do with elastase inhibition.

3.       This is further supported by the loss of protection when sivelestat is given later on (5h after APAP), when its capacity to interfere with the onset of APAP metabolism/injury is likely gone. However, at that time the neutrophil-induced injury phase would likely start, which means that sivelestat should still protect if neutrophil elastase would be the actual target. Consistent with this paper, Marques et al 2021 (https://pubmed.ncbi.nlm.nih.gov/34532999/) showed that elastase inhibitors to not reduce liver injury when given 6h post APAP.

4.       In general, the inflammatory response after APAP overdose is a sterile inflammatory response. This means that DAMPs released by severe APAP-induced necrosis trigger inflammatory mediator production, which recruit neutrophils into the liver. Thus, extensive necrosis has to occur before neutrophil accumulation in the liver, which is not the case as reported by the authors. Second, in case of a sterile inflammatory response, specific interventions targeted against the inflammatory response could at best attenuate the injury caused by neutrophils but not the initiating necrosis caused by APAP metabolism and cytotoxicity. However, as the authors show in Figure 2, sivelestat eliminates also the initial injury. This is inconsistent with an effect on inflammation only. Thus, the conclusions are not supported by the data.

Reviewer 4 Report

In this article, the authors investigated the efficacy of a potent and specific neutrophil elastase inhibitor (sivelestat) in mice with acetaminophen-induced acute liver injury in order to elucidate the involvement of neutrophil elastase in acetaminophen-induced liver injury. BALB/c mice was used for the establishment of APAP induced liver injury model and the efficacy of serum transaminase, iNOS expression and histopathology was used for evaluation.

 1. In the discussion section, “We previously demonstrated that 750 mg/kg APAP is a lethal dose for BALB/c mice, as approximately 50% of the mice died after administration of 750 mg/kg APAP”. Dose the author means “a lethal dose” or “median lethal dose (LD50)” of APAP for BALB/c mice?

2. AST is also an important indicator of liver injury, the author should provide the results of AST.

3.  Why does 3 and 5 hours were chosen for giving sivelestat? Please provide evidence. Besides, the author said that “In the PBS-treated group, 50% of the mice (6 deaths/12 mice) succumbed to acute liver injury within 24 h of the challenge”. Please provide detailed data (such as how many death during XX hours) so that the chosen time point could be well-founded.

4.  Please make sure that the format of the manuscript should be consistent with the rules, such as a space or no space at the beginning of a paragraph should be consist.

5.  There are many grammatical errors in the article, so it is suggested that the language need to be improved.

Author Response

Reviewer 4

  1. In the discussion section, “We previously demonstrated that 750 mg/kg APAP is a lethal dose for BALB/c mice, as approximately 50% of the mice died after administration of 750 mg/kg APAP”. Dose the author means “a lethal dose” or “median lethal dose (LD50)” of APAP for BALB/c mice?

It means “a lethal dose”, not LD50.

  1. AST is also an important indicator of liver injury, the author should provide the results of AST.

In accordance with the comment, data on serum AST levels over time were added to Figure 2b. In addition, some text was revised (lines 98-99, 103-104, 287-288, 290, 624).

  1. Why does 3 and 5 hours were chosen for giving sivelestat? Please provide evidence. Besides, the author said that “In the PBS-treated group, 50% of the mice (6 deaths/12 mice) succumbed to acute liver injury within 24 h of the challenge”. Please provide detailed data (such as how many deaths during XX hours) so that the chosen time point could be well-founded.

Since myeloperoxidase (MPO) activity increased significantly 6 h after APAP administration (Fig. 3c), and it is possible that the key events leading to hepatotoxicity had been completed by this time, we considered that administration of sivelestat after 6 h of APAP administration may not exert its protective effect by modulating neutrophil elastase. In addition, because we considered therapeutic administration to be important, we treated sivelestat at 3 and 5 h after APAP administration and analyzed its therapeutic effect.

In accordance with the comments, the sentence was modified as “In the PBS-treated group, 50% of the mice (4 animals died within 12 h, 2 within 24 h, total 6/12 animals died within 24 h) succumbed to acute liver injury within 24 h of the challenge (line 85-87)”. Please see Figure 1a and attached Excel data for survival.

  1. Please make sure that the format of the manuscript should be consistent with the rules, such as a space or no space at the beginning of a paragraph should be consist.

In accordance with the comments, the format was modified according to IJMS-template.

  1. There are many grammatical errors in the article, so it is suggested that the language need to be improved.

In accordance with the comments, our manuscript was carefully edited in English by an English speaker (please see the attached certification).

Round 2

Reviewer 2 Report

The authors replied correctly to the reviewer's comments. 

Author Response

Reviewer 2

The authors replied correctly to the reviewer's comments.

Thank you for your constructive and valuable suggestions.

Reviewer 4 Report

During the last round we asked the following question:

In the discussion section, “We previously demonstrated that 750 mg/kg APAP is a lethal dose for BALB/c mice, as approximately 50% of the mice died after administration of 750 mg/kg APAP”. Dose the author means “a lethal dose” or “median lethal dose (LD50)” of APAP for BALB/c mice?

The author seems to have given the wrong answer. LD50 means the dose of  50% mice death, not  “a lethal dose” . Please correct it.

In addition, we noticed that the first author of the article seems to have changed, but the research content has not changed. Please explain.

Author Response

Reviewer 4

During the last round we asked the following question:

In the discussion section, “We previously demonstrated that 750 mg/kg APAP is a lethal dose for BALB/c mice, as approximately 50% of the mice died after administration of 750 mg/kg APAP”. Dose the author means “a lethal dose” or “median lethal dose (LD50)” of APAP for BALB/c mice?

The author seems to have given the wrong answer. LD50 means the dose of 50% mice death, not “a lethal dose”. Please correct it.

The main aim of our previous study was not to determine the APAP dose. In comparison between BALB/c mice and IFN-gamma KO one, we could get most demonstrable phenotype in survival rate in the use of 750 mg/kg APAP. Moreover, our other study demonstrated that APAP hepatotoxicity was different dependently on mouse genetic strain. For example, the LD50 of APAP in C57BL/6 mice was lower than that in BALB/c mice. Thus, we mentioned “a lethal dose” in our manuscript. In accordance with the comment, we corrected the manuscript (lane 240).

In addition, we noticed that the first author of the article seems to have changed, but the research content has not changed. Please explain.

In the first version of manuscript, Siying Zhang and Yuko Ishida contributed equally and sheared the first authorship. In the revision process, Reviewer 2 requested us to perform in vitro experiments using intraperitoneal macrophages but not a macrophage cell line (RAW264.7), and Reviewer 3 also requested us to show the presence of neutrophils in the liver of mice before APAP challenge. In order to respond to these requests , all of these additional experiments were performed by Yuko Ishida. Subsequently, she rewrote the manuscript to accommodate all reviewers with me. Before submission of revised version, we discussed about the final authorship in this manuscript, and we reached to an agreement that Yuko Ishida would be designated as the appropriate first author of this study.